# The Flavonoid Biosynthesis Network in Plants

**DOI:** 10.3390/ijms222312824

**Published:** 2021-11-26

**Authors:** Weixin Liu, Yi Feng, Suhang Yu, Zhengqi Fan, Xinlei Li, Jiyuan Li, Hengfu Yin

**Affiliations:** 1State Key Laboratory of Tree Genetics and Breeding, Research Institute of Subtropical Forestry, Chinese Academy of Forestry, Hangzhou 311400, China; lwx060624@163.com (W.L.); fy11071107@163.com (Y.F.); yusuhang819@163.com (S.Y.); fzq_76@126.com (Z.F.); lixinlei2020@163.com (X.L.); 2Key Laboratory of Forest Genetics and Breeding, Research Institute of Subtropical Forestry, Chinese Academy of Forestry, Hangzhou 311400, China

**Keywords:** flavonoids, biosynthesis, molecular structure, biosynthetic enzyme, gene regulation

## Abstract

Flavonoids are an important class of secondary metabolites widely found in plants, contributing to plant growth and development and having prominent applications in food and medicine. The biosynthesis of flavonoids has long been the focus of intense research in plant biology. Flavonoids are derived from the phenylpropanoid metabolic pathway, and have a basic structure that comprises a C15 benzene ring structure of C6-C3-C6. Over recent decades, a considerable number of studies have been directed at elucidating the mechanisms involved in flavonoid biosynthesis in plants. In this review, we systematically summarize the flavonoid biosynthetic pathway. We further assemble an exhaustive map of flavonoid biosynthesis in plants comprising eight branches (stilbene, aurone, flavone, isoflavone, flavonol, phlobaphene, proanthocyanidin, and anthocyanin biosynthesis) and four important intermediate metabolites (chalcone, flavanone, dihydroflavonol, and leucoanthocyanidin). This review affords a comprehensive overview of the current knowledge regarding flavonoid biosynthesis, and provides the theoretical basis for further elucidating the pathways involved in the biosynthesis of flavonoids, which will aid in better understanding their functions and potential uses.

## 1. Introduction

Flavonoids comprise a group of phenylpropanoids that, as water-soluble pigments, are stored in the vacuoles of plant cells [1]. Except for stilbenes (a class of flavonoids), which has a C6-C2-C6 structure (Figure 1), the basic structure of flavonoids consists of a C6-C3-C6 carbon skeleton (Figure 1) comprising two 6-carbon benzene rings (rings A and B) linked by a 3-carbon heterocyclic ring (ring C) [2]. Flavonoids can be classified into 12 subgroups—chalcones, stilbenes, aurones, flavanones, flavones, isoflavones, phlobaphenes, dihydroflavonols, flavonols, leucoanthocyanidins, proanthocyanidins, and anthocyanins (Figure 1) [3,4]—based on the degree of oxidation of the heterocyclic ring and the number of hydroxyl or methyl groups on the benzene ring. At the same time, various modifications (glycosylation, acylation, and others) and molecular polymerization lead to the formation of a large number of flavonoid compounds [5,6]. To date, more than 9000 plant flavonoids have been isolated and identified [7]. 

Some flavonoids play an important role in plant development and defense. Flavonoids constitute one of the main pigments in plants, such as anthocyanins (red, orange, blue, and purple pigments); chalcones and aurones (yellow pigments); and flavonols and flavones (white and pale-yellow pigments), which impart on plants a wide variety of colors [8]. Flavonoids, as phytoalexins or antioxidants, have reactive oxygen species (ROS) scavenging ability [9] and protect plants against damage from biotic and abiotic stresses, including UV irradiation, cold stress, pathogen infection, and insect feeding [10,11,12]. In plants, flavonoids can also act as signaling molecules, attracting insects for pollination and participating in auxin metabolism [13]. Plant flavonoids also have widespread use in daily life, such as for food and medicinal purposes. For instance, anthocyanins and proanthocyanidins are important edible pigments and taste-regulating components in food and wine [4], while plant flavonoids, administered as active ingredients, can help delay the aging of the nervous system, immune organs, reproductive system, liver, and skin, and also contribute to the prevention of osteoporosis, cardiovascular disease, Alzheimer’s disease, and breast cancer [14,15,16].

Flavonoids have long been a major focus of research into secondary metabolism. On PubMed, performing a search using ‘flavonoid’ as a search term retrieves more than 10,000 articles in both 2019 and 2020. Recent decades have witnessed a considerable renewed interest in flavonoid biosynthesis in plants. In this review, we present a systematic summary of what is known of the flavonoid biosynthetic pathway in plants, presenting a model of flavonoid biosynthesis that includes eight branches and four intermediate metabolites (Figure 2), thereby providing a theoretical basis for the genetic improvement of flavonoid metabolism as well as improving our understanding of their functions and potential uses.

## 2. Flavonoid Biosynthesis in Plants

### 2.1. The General Phenylpropanoid Pathway

Flavonoids are generated from phenylalanine through the phenylpropanoid pathway, while phenylalanine is synthesized via the shikimate pathway [17]. The first three steps in the phenylpropanoid pathway are referred to as the general phenylpropanoid pathway [1]. In this pathway, phenylalanine, an aromatic amino acid, is converted to *p*-coumaroyl-CoA through the activity of phenylalanine ammonia lyase (PAL), cinnamic acid 4-hydroxylase (C4H), and 4-coumarate: CoA ligase (4CL). PAL catalyzes the first committed step in the general phenylpropanoid pathway, namely, the deamination of phenylalanine to *trans*-cinnamic acid [18]. Additionally, PAL plays a key role in mediating carbon flux from primary to secondary metabolism in plants [19]. PAL activity has been linked to the concentrations of anthocyanins and other phenolic compounds in strawberry fruit [20] while *StlA*, a *Photorhabdus luminescens* PAL-encoding gene, was shown to be involved in the production of a stilbene antibiotic [18]. The second step in the general phenylpropanoid pathway involves the activity of C4H, a cytochrome P450 monooxygenase in plants, which catalyzes the hydroxylation of *trans*-cinnamic acid to generate *p*-coumaric acid. This is also the first oxidation reaction in the flavonoid synthesis pathway [21]. In *Populus trichocarpa* and *Arabidopsis thaliana*, the expression level of *C4H* has been associated with the content of lignin, an important phenylpropanoid metabolite [1]. In the third step of the general phenylpropanoid pathway, 4CL catalyzes the formation of *p*-coumaroyl-CoA by the addition of a co-enzyme A (CoA) unit to *p*-coumaric acid. In plants, the *4CL* gene usually exists as a family the members of which mostly display substrate specificity. Of the four *4CL* genes in *A. thaliana*, *At4CL1*, *At4CL2*, and *At4CL4* are involved in lignin biosynthesis, while *At4CL3* has a role in flavonoid metabolism [22]. In plants, the activity of 4CL is positively correlated with the anthocyanin and flavonol content in response to stress [23], while *PAL*, *C4H*, and *4CL* are often coordinately expressed [24]. The general phenylpropanoid pathway is common to all the downstream metabolites, such as flavonoids and lignin. In this review, we focus on the flavonoid biosynthetic pathway, and present a model that includes eight branches—the biosynthesis of stilbenes, aurones, flavones, isoflavones, flavonols, phlobaphenes, proanthocyanidins, and anthocyanins—and four important intermediate metabolites, namely, chalcones, flavanones, dihydroflavonols, and leucoanthocyanidins (Figure 2).

### 2.2. Chalcone: The First Key Intermediate Metabolite in Flavonoid Biosynthesis

The entry of *p*-coumaroyl-CoA into the flavonoid biosynthesis pathway represents the start of the synthesis of specific flavonoids, which begins with chalcone formation [2]. One molecule of *p*-coumaroyl-CoA and three molecules of malonyl-COA, derived from acetyl-CoA via the activity of acetyl-CoA carboxylase (ACCase), generate naringenin chalcone (4,2′,4′,6′-tetrahydroxychalcone [THC] [chalcone]) through the action of chalcone synthase (CHS) [25]. CHS, a polyketide synthase, is the key and first rate-limiting enzyme in the flavonoid biosynthetic pathway [26,27]. In tomato (*Solanum lycopersicum*), RNA interference (RNAi)-mediated suppression of *CHS* leads to a reduction in total flavonoid levels [28]. Chalcone reductase (CHR), an aldo-keto reductase superfamily member, acts on an intermediate of the CHS reaction, catalyzing its C-6′ dehydroxylation, yielding isoliquiritigenin (4,2′,4′-trihydroxychalcone [deoxychalcone]) [29]. Overexpressing the *CHR1* gene from *Lotus japonicus* in petunia leads to the formation of isoliquiritigenin and a decrease in anthocyanin content [30]. Because THC is readily converted to a colorless naringenin under the action of chalcone isomerase (CHI) or through spontaneous isomerization, it is frequently converted to the more stable THC 2′-glucoside (isosalipurposide [ISP]) under the action of chalcone 2′-glucosyltransferase (CH2′GT) in plant vacuoles [31,32]. Differences in CH2′GT gene expression or enzymatic activity might account for the difference in ISP content in the petals of different varieties of yellow carnation [33]. Chalcone is the first key intermediate product in the flavonoid metabolic pathway, providing a basic skeleton for downstream flavonoid synthesis. Chalcone (THC, isoliquiritigenin, and ISP, among others) is also an important yellow pigment in plants [31].

### 2.3. Stilbene Biosynthesis: The First Branch of the Flavonoid Biosynthesis Pathway

Stilbene synthase (STS) also uses *p*-coumaroyl-CoA and malonyl-CoA as substrates and catalyzes the formation of the stilbene backbone, such as resveratrol [34,35]. The stilbene pathway is the first branch of the flavonoid biosynthesis pathway and exists only in a few plants, such as grapevine, pine, sorghum, and peanut [36,37]. STS, a member of the type III polyketide synthase family, is the first and key enzyme in stilbene biosynthesis, and is closely related to, and evolved from, CHS [34]. However, STS generates a compound with a different C14 backbone (C6-C2-C6) along with the release of 4 carbon dioxide (CO_2_) molecules, while CHS catalyzes the formation of C15 skeletons (C6-C3-C6), with only 3 molecules of CO_2_ being released [38]. In *Vitis amurensis* calli, the overexpression of *Picea jezoensis PjSTS1a*, *PjSTS2*, and *PjSTS3* greatly increases the total stilbene content [39]. Most plant stilbenes are derivatives of the basic unit *trans*-resveratrol (3,5,4′-trihydroxy-*trans*-stilbene) that has undergone various modifications, such as isomerization, glycosylation, methylation, oligomerization, and prenylation [36]. *Trans*-resveratrol can be converted to polydatin, pterostilbene, and piceatannol by glycosylation, methylation, and hydroxylation, respectively [35]. In peanuts, the major prenylated stilbene compounds are *trans*-3′-(3-methyl-2-butenyl)-resveratrol and *trans*-arachidin-1/2/3 [40]. Viniferin and *cis*-stilbene are derived from the oligomerization and isomerization of *trans*-resveratrol, respectively [36,41].

### 2.4. Aurone Biosynthesis: The Bright Yellow Pigment Pathway

Aurones, important yellow pigments in plants, comprise a class of flavonoids derived from chalcone [42]. Aurone pigments produce brighter yellow coloration than chalcones and are responsible for the golden color in some popular ornamental plants [31]. Aurones are found in relatively few plant species, such as snapdragon, sunflowers, and coreopsis [42,43]. THC is the direct substrate for aurone biosynthesis [44]. First, chalcone 4′-*O*-glucosyltransferase (CH4′GT) catalyzes the formation of THC 4′-*O*-glucoside from THC in the plant cytoplasm. The former is then transferred to the vacuole and converted to aureusidin 6-*O*-glucoside (aurone) by the action of aureusidin synthase (AS) [45,46]. AS can also catalyze the formation of aureusidin directly from THC; aureusidin and its glycosides are the main pigments in the yellow petal of *Antirrhinum majus* and *Dahlia variabilis* [47]. 2′,4′,6′,3,4-Pentahydroxychalcone (PHC, a type of chalcone) can also be converted into aurones (bracteatin and bracteatin 6-*O*-glucoside) by CH4′GT and/or AS [31,47]. CH4′GT and CHI can both use chalcone as a substrate, and 4′-gulcosylation by CH4′GT not only provides a direct precursor for aurone synthesis, but also inhibits the isomerization activity of CHI by repressing key interactions between CHI and the 4′-hydroxy group of chalcones [48]. AS, a homolog of plant polyphenol oxidase (PPO), catalyzes the 4-monohydroxylation or 3,4-dihydroxylation of ring B to produce aurone, followed by oxidative cyclization by oxygenation [49]. Both in *Ipomoea nil* [50] and *Torenia* [45], the co-overexpression of the *AmCH4′GT* and *AmAS1* genes leads to the accumulation of aurone 6-*O*-glucoside. Furthermore, various classical substitution patterns, such as hydroxylation, methoxylation, and glycosylation, lead to the formation of a series of aurone compounds, with over 100 structures having been reported to date [48].

### 2.5. Flavanones: The Central Branch Point in the Flavonoid Biosynthesis Pathway

CHI catalyzes the intramolecular cyclization of chalcones to form flavanones in the cytoplasm, resulting in the formation of the heterocyclic ring C in the flavonoid pathway [2,51]. In general, CHIs can be classified into two types in plants according to the substrate utilized [52]. Type I CHIs, ubiquitous in vascular plants, are responsible for the conversion of THC into naringenin [53]. Type II CHIs are found primarily in leguminous plants and can utilize either THC or isoliquiritigenin to generate naringenin and liquiritigenin [1]. Apart from these two types, two other types of CHI exist (type III and type IV), which retain the catalytic activity of the CHI fold but not chalcone cyclization activity [54]. In bacteria, some CHI-like enzymes catalyze a reversible reaction in the flavonoid pathway that converts flavanones to chalcones [8]. CHI is the second key rate-limiting enzyme in the flavonoid biosynthesis pathway [52]. The expression level of *CHI* was found to be positively correlated with flavonoid content in *A. thaliana* [55]. In both *Dracaena cambodiana* and tobacco, the overexpression of *DcCHI1* or *DcCHI4* leads to increased flavonoid accumulation [53]. In transgenic tobacco plants, RNAi-mediated suppression of *CHI* enhances the level of chalcone in pollen [56]. Furthermore, naringenin can be converted to eriodictyol and pentahydroxyflavanone (two flavanones) under the action of flavanone 3′-hydroxylase (F3′H) and flavanone 3′,5′-hydroxylase (F3′5′H) at position C-3 and/or C-5 of ring B [8]. Flavanones (naringenin, liquiritigenin, pentahydroxyflavanone, and eriodictyol) represent the central branch point in the flavonoid biosynthesis pathway, acting as common substrates for the flavone, isoflavone, and phlobaphene branches, as well as the downstream flavonoid pathway [51,57].

### 2.6. Flavone Biosynthesis

Flavone biosynthesis is an important branch of the flavonoid pathway in all higher plants. Flavones are produced from flavanones by flavone synthase (FNS); for instance, naringenin, liquiritigenin, eriodictyol, and pentahydroxyflavanone can be converted to apigenin, dihydroxyflavone, luteolin, and tricetin, respectively [58,59,60]. FNS catalyzes the formation of a double bond between position C-2 and C-3 of ring C in flavanones and can be divided into two classes—FNSI and FNSII [61]. FNSIs are soluble 2-oxoglutarate- and Fe^2+^-dependent dioxygenases mainly found in members of the Apiaceae [62]. Meanwhile, FNSII members belong to the NADPH- and oxygen-dependent cytochrome P450 membrane-bound monooxygenases and are widely distributed in higher plants [63,64]. FNS is the key enzyme in flavone formation. *Morus notabilis FNSI* can use both naringenin and eriodictyol as substrates to generate the corresponding flavones [62]. In *A. thaliana*, the overexpression of *Pohlia nutans FNSI* results in apigenin accumulation [65]. The expression levels of *FNSII* were reported to be consistent with flavone accumulation patterns in the flower buds of *Lonicera japonica* [61]. In *Medicago truncatula*, meanwhile, *MtFNSII* can act on flavanones, generating intermediate 2-hydroxyflavanones (instead of flavones), which are then further converted into flavones [66]. Flavanones can also be converted to *C*-glycosyl flavones (Dong and Lin, 2020). Naringenin and eriodictyol are converted to apigenin *C*-glycosides and luteolin *C*-glycosides under the action of flavanone-2-hydroxylase (F2H), *C*-glycosyltransferase (CGT), and dehydratase [67].

*Scutellaria baicalensis* is a traditional medicinal plant in China and is rich in flavones such as wogonin and baicalein [17]. There are two flavone synthetic pathways in *S. baicalensis*, namely, the general flavone pathway, which is active in aerial parts; and a root-specific flavone pathway [68]), which evolved from the former [69]. In this pathway, cinnamic acid is first directly converted to cinnamoyl-CoA by cinnamate-CoA ligase (SbCLL-7) independently of C4H and 4CL enzyme activity [70]. Subsequently, cinnamoyl-CoA is continuously acted on by CHS, CHI, and FNSII to produce chrysin, a root-specific flavone [69]. Chrysin can further be converted to baicalein and norwogonin (two root-specific flavones) under the catalysis of respectively flavonoid 6-hydroxylase (F6H) and flavonoid 8-hydroxylase (F8H), two CYP450 enzymes [71]. Norwogonin can also be converted to other root-specific flavones—wogonin, isowogonin, and moslosooflavone—under the activity of *O*-methyl transferases (OMTs) [72]. Additionally, F6H can generate scutellarein from apigenin [70]. The above flavones can be further modified to generate additional flavone derivatives.

### 2.7. Isoflavone Biosynthesis

The isoflavone biosynthesis pathway is mainly distributed in leguminous plants [73]. Isoflavone synthase (IFS) leads flavanone to the isoflavone pathway [74] and appears to be able to use both naringenin and liquiritigenin as substrates to generate 2-hydroxy-2,3-dihydrogenistein and 2,7,4′-trihydroxyisoflavanone, respectively [75,76]. These are further converted to isoflavone genistein and daidzein under the action of hydroxyisoflavanone dehydratase (HID) [77]. Liquiritigenin can also be first converted to 6,7,4′-trihydroxyflavanone by F6H, and then to glycitein (an isoflavone) through the catalytic activities of IFS, HID, and isoflavanone *O*-methyl transferase (IOMT) [78]. IFS and HID catalyze two reactions to produce isoflavone, that is, the formation of a double bond between positions C-2 and C-3 of ring C and a shift of ring B from position C-2 to C-3 of ring C [79,80]. IFS, a cytochrome P450 hydroxylase, is the first and key enzyme in the isoflavone biosynthesis pathway [81]. The overexpression of *Glycine max IFS* in *Allium cepa* led to the accumulation of the isoflavone genistein in in vitro tissues [82]. Knocking out the expression of the *IFS1* gene using CRISPR/Cas9 led to a significant reduction in the levels of isoflavones such as genistein [58]. Various modifications further generate specific isoflavones. Daidzein is converted to puerarin or formononetin by a specific glycosyltransferase (GT) or IOMT [79,83]. Malonyltransferase (MT) can act on isoflavones (genistein, daidzein, and glycitein) to generate the corresponding malonyl-isoflavones (malonylgenistein, malonyldaidzein, and malonylglycitein) [80]. Moreover, the successive enzymatic reactions catalyzed by IOMT, isoflavone reductase (IFR), isoflavone 2′-hydroxylase (I2′H) or isoflavone 3′-hydroxylase (I3′H), vestitone reductase (VR), pterocarpan synthase (PTS), and 7,2′-dihydroxy-4′-methoxyisoflavanol dehydratase (DMID) lead to the accumulation of isoflavonoids such as maackiain and pterocarpan [1,84,85].

### 2.8. Phlobaphene Biosynthesis

Besides flavones and isoflavones, the biosynthesis of phlobaphenes also uses flavanones as substrates [86]. Phlobaphenes are reddish insoluble pigments in plants [87] and are predominantly found in seed pericarp, cob-glumes, tassel glumes, husk, and floral structures of plants such as maize and sorghum [88,89,90]. Flavanone 4-reductase (FNR) acts on flavanones (naringenin and eriodictyol) to form the corresponding flanvan-4-ols (apiforol and luteoforol), which are the immediate precursors of pholbaphenes [91,92]. Apiforol and luteoforol are then further polymerized to generate phlobaphenes [57]. FNR is a NADPH-dependent reductase and drives the substitution of an oxygen with a hydroxyl group at position C-4 of ring C [89]. FNR is also a dihydroflavonol 4-reductase (DFR)-like enzyme, and can convert dihydroflavonol to leucoanthocyanidin [93]. In maize, DFR and FNR correspond to the same enzyme [91]. The inhibition of flavanone 3-hydroxylase (F3H) activity promotes the conversion of flavanone to flavan-4-ol through the catalytic activity of FNR in *Sinningia cardinalis* and *Zea mays* [94].

### 2.9. Dihydroflavonol: A Key Branch Point in the Flavonoid Biosynthesis Pathway

Dihydroflavonol (or flavanonol) is an important intermediate metabolite and a key branch point in the flavonoid biosynthesis pathway. Dihydroflavonol is generated from flavanone under the catalysis of F3H and is the common precursor for flavonol, anthocyanin, and proanthocyanin [95,96]. F3H acts on naringenin, eriodictyol, and pentahydroxyflavanone to form the corresponding dihydroflavonols, namely, dihydrokaempferol (DHK), dihydroquercetin (DHQ), and dihydromyricetin (DHM) [97,98]. Moreover, DHK can be converted to DHQ by F3′H and DHK, while DHQ can generate DHM under the action of F3′5′H [51]. 

F3H, a FeⅡ/2-oxoglutarate-dependent dioxygenase, catalyzes the dydroxylation of flavonones at position C-3 and is the key enzyme in dihydroflavonol synthesis [99]. Because flavanones are also the substrates in the flavone, isoflavone, and phlobaphene biosynthetic pathways, F3H competes with FNS, IFS, and FNR for these common substrates [98]. The overexpression of *F3H* leads to the generation of DHK in tobacco and yeast [100]. In *Silybum marianum*, F3H was shown to catalyze the synthesis of taxifolin (DHQ) from eriodictyol [101], while the expression of *AgF3H* was significantly positively correlated with DHM content in different tissues of *Ampelopsis grossedentata* [102].

F3′H and F3′5′H, both cytochrome P450 enzymes, catalyze the hydroxylation of flavonoids at position C-3′ or C-3′ and C-5′ of ring B, respectively, so as to the formation of substrates of different pathways [8,103]. F3′H and F3′5′H generate flavanones with differing degrees of hydroxylation, resulting in naringenin, eriodictyol, and pentahydroxyflavanone entering different flavone synthetic pathways [60]. F3′H catalyzes the production of DHQ, which is the synthetic precursor of cyanidin in the anthocyanidin pathway and quercetin in the flavonol pathway [104]. DHM, synthesized by F3′5′H, is the direct precursor of delphinidin in the anthocyanidin pathway and myricetin in the flavonol pathway, while DHK can be converted to pelargonidin (an anthocyanidin) and kaempferol (a flavonol) [3,98]. Thus, F3′H and F3′5′H are the determinants of flavonoid composition in many plants and the key enzymes in flavonoid biosynthesis. The ectopic expression of apple F3′H genes increases the levels of quercetin and cyanidin in Arabidopsis and tobacco [105]. Meanwhile, delphinidin levels are decreased while those of cyanidin are increased in a natural *Glycine soja f3′5′h* mutant [106].

### 2.10. Flavonol Biosynthesis

Flavonols are flavonoid metabolites that are hydroxylated at position C-3 of ring C [51]. Their C-3 position is highly prone to glycosidation; accordingly, they often exist in plant cells in glycosidated forms [98]. The dihydroflavonols DHK, DHQ, and DHM are respectively converted to the flavonols kaempferol, quercetin, and myricetin by flavonol synthase (FLS) [107]. F3′H can also catalyze the conversion of kaempferol to quercetin, while F3′5′H activity generates myricetin from kaempferol or quercetin [108]. Kaempferol, quercetin, and myricetin are further modified to various flavonol derivatives through the activities of enzymes such as methyl transferases, GTs, and acyltransferase (AT), among others [60,109]. FLS, a FeⅡ/2-oxoglutarate-dependent dioxygenase, is the key and rate-limiting enzyme in the flavonol biosynthesis pathway [110] and catalyzes the desaturation of dihydroflavonol to form a C-2 and C-3 double bond in ring C [111]. The ectopic expression of *Camellia sinensis FLSa/b/c* in tobacco promoted the accumulation of kaempferol and a decrease in anthocyanin content in flowers [112]. Meanwhile, the overexpression of *FLS* of *Allium cepa* in tobacco enhanced quercetin signals in the roots [113].

### 2.11. Leucoanthocyanidin and Anthocyanin Biosynthesis

DFR, a NADPH-dependent reductase, is the key enzyme in flavonoid metabolism in the anthocyanidin and proanthocyanidin pathway and catalyzes the formation of a hydroxyl group at position C-4 of ring C [114,115,116]. DFR catalyzes the reduction of dihydroflavonols, DHK, DHQ, and DHM to produce their respective leucoanthocyanidins (also known as flavan-3,4-ols or flavan-diols), leucopelargonidin, leucocyanidin, and leucodelphinidin [117]. Because DHK, DHQ, and DHM are very similar in structure, differing only in the numbers of hydroxyl groups on the B ring (which is not the site of enzymatic action), DFR can use all three as substrates in most plants [118]. In *Vitis vinifera*, DFR converts DHK to leucopelargonidin [119]. The overexpression of *Brassica oleracea DRF1* was shown to promote anthocyanin accumulation, whereas the virus-induced silencing of the *BoDRF1* gene elicited the opposite effect [120]. DFR and FLS compete for common dihydroflavonol substrates and *DFR* and *FLS* inhibit each other’s transcription [121].

Leucoanthocyanidin is an important intermediate product in the flavonoid pathway and the direct synthetic precursor of anthocyanidin and proanthocyanidin. The colorless leucopelargonidin, leucocyanidin, and leucodelphinidin are transformed into the corresponding anthocyanidins (the colored pelargonidin, cyanidin, and delphinidin) under the catalysis of anthocyanidin synthase (ANS), also known as leucoanthocyanidin dioxygenase (LDOX) [122,123]. Like FLS, F3H, and FNSI, ANS/LDOX is also a FeⅡ/2-oxoglutarate-dependent dioxygenase, and catalyzes the dehydroxylation of C-4 and the formation of a double bond in ring C [3,124]. *ANS* overexpression in strawberry enhanced the anthocyanin concentration [125]. In plants, unstable anthocyanidins are converted to stable anthocyanins, namely, pelargonidin-3-glucoside, cyanidin-3-glucoside, and delphinidin-3-glucoside, by UDP-glucose flavonoid 3-glucosyltransferase (UFGT) [126,127]. OMT can further catalyze the conversion of cyanidin-3-glucoside to peonidin glycoside and delpinidin-3-glucoside to petunidin glycoside or malvidin glycoside [118,128]. The pelargonidin, cyanidin, peonidin, delphinidin, petunidin, and malvidin glycosides constitute six major categories of anthocyanins and their further modifications (acylation, glycosylation, and methylation) lead to the formation of various anthocyanins [5,118,127].

In addition to the above-mentioned anthocyanins, a rare type of anthocyanin, 3-deoxyanthocyanidin, also exists in plants [129]. The biosynthesis of 3-deoxyanthocyanidins is similar to that of the phlobaphenes, and they both use flavan-4-ols (luteoferol and apiferol) as substrates [130]. Luteoforol and apiferol are respectively transformed into 3-deoxyanthocyanidins (luteolinidin and apigeninidin) by an unknown enzyme, likely with anthocyanidin synthase-like activity [57,94]. Luteolinidin and apigeninidin are further converted into 3-deoxyanthocyanidin glycosides (3-deoxyanthocyanins) by GT [131]. Unlike anthocyanidins, 3-deoxyanthocyanidins lack a hydroxyl group at position C-3 of ring C, giving them greater stability under temperature fluctuations as well as greater color stability [132,133]. In plants, 3-deoxyanthocyanidins mainly exist in the aglycone form, and not as 3-deoxyanthocyanins, whereas the anthocyanidins primarily exist in glycoside form (anthocyanins) [133]. 3-Deoxyanthocyanidins have been found in many plants, including sorghum and maize [130,134]; however, their biosynthetic pathway needs to be further analyzed.

### 2.12. Proanthocyanidin Biosynthesis

Proanthocyanidins, also known as condensed tannins, are an important type of flavonoid synthesized from leucoanthocyanidins and anthocyanidins. Leucoanthocyanidin reductase (LAR) converts leucoanthocyanidins, leucopelargonidin, leucocyanidin, and leucodelphinidin to *trans*-flavan-3-ols, afzelechin, catechin, and gallocatechin, respectively [135,136]. LAR, a NADPH-dependent reductase, drives the C-4 dehydroxylation of the C ring [137]. Anthocyanidin reductase (ANR) can convert anthocyanidins, pelargonidin, cyanidin, and delphinidin, into the corresponding *cis*-flavan-3-ols, epiafzelechin, epicatechin, and epigallocatechin [138]. ANR is also a NADPH-dependent reductase and catalyzes the removal of a double bond at ring C [139]. Flavan-3-ols, *trans*-flavan-3-ols, and *cis*-flavan-3-ols are the basic proanthocyanidin units. Proanthocyanidins are synthesized via the polymerization (or condensation) of flavan-3-ols [140,141]. Colorless proanthocyanidins are transferred to plant vacuoles [142] and can be oxidized to generate colored tannins (yellow to brown) by polyphenol oxidase (PPO) [135]. LAR and ANR are the key and rate-limiting enzymes in proanthocyanidin biosynthesis. In *Populus tomentosa*, the overexpression of *LAR3* greatly increased the proanthocyanidin levels [143]. The ectopic expression of *OvBAN*, an *ANR* gene from *Onobrychis viviaefolia*, in alfalfa (*Medicago sativa*) promoted ANR enzyme activity and enhanced proanthocyanidin content [144]. Because they use the same substrates, a competitive relationship exists between the proanthocyanidin and anthocyanin biosynthesis pathways [145].

## 3. Transcriptional Regulation of Flavonoid Biosynthesis in Plants

Transcriptional control plays a central role in the modulation of flavonoid biosynthesis (Figure 3). The MBW complex, composed of MYB, bHLH, and WD40, is the main transcriptional regulator in flavonoid biosynthesis [146]. MYB transcription factors have a conserved MYB domain in the N-terminus that is required for DNA binding and interaction with other proteins [147]. MYB proteins can be divided into four groups—1R-MYB/MYB-related, R2R3-MYB, 3R-MYB, and 4R-MYB—according to the number and position of MYB domain repeats [117]. Members of the R2R3-MYB group are mainly involved in regulating flavonoid metabolism [148]. The overexpression of *AN4* (a R2R3-MYB-encoding gene) can enhance anthocyanin biosynthesis by promoting the expression of anthocyanin biosynthesis genes, such as *CHS*, *CHI*, *F3H*, and *DFR* [149]. In *Cucumis sativus*, the R2R3-MYB transcription factor CsMYB60 induced the expression of CsFLS and CsLAR by binding to their promoters, thereby promoting flavonol and proanthocyanidin biosynthesis [150]. MYB transcription factors also act as repressors in the regulation of flavonoid biosynthesis. For instance, in the apple (*Malus domestica*), MdMYB15L was reported to interact with MdbHLH33 and inhibit the promotion of the MdbHLH33-MYB-WD40 (MBW) complex, thereby also suppressing anthocyanin biosynthesis [151].

bHLH transcription factors have been shown to participate in the regulation of flavonoid biosynthesis. The transient expression of *DhbHLH1* induces anthocyanin synthesis in the white petals of *Dendrobium* hybrids [152]. In *Dianthus caryophyllus*, meanwhile, the “red speckles and stripes on white petals” phenotype results from the local expression of *bHLH*, which promotes the expression of DFR and that of downstream enzymes in the anthocyanin biosynthetic pathway [153].

WD40, widely present in eukaryotic cells, contains multiple tandem repeats of a WD motif and interacts with other proteins through its WD domain [1]. Generally, WD40 does not directly bind to target gene promoters, forming instead a complex with MYB and bHLH in the regulation of flavonoid biosynthesis. The WD40 protein TTG1 regulated anthocyanin metabolism through MYB/bHLH/TTG1 complex [154]. Moreover, in tomato, the WD40 protein SlAN11 was shown to induce anthocyanin and proanthocyanidin biosynthesis and limit flavonol accumulation by repressing *FLS* expression [155].

Also in tomato, besides the MBW complex, the transcription factors NF-YA, NF-YB, and NF-YC can reportedly form a NF-Y protein complex that binds to the promoter of the *CHS1* gene, thereby regulating flavonoid synthesis and affecting tomato peel color [25]. Additionally, the ethylene response factors Pp4ERF24 and Pp12ERF96, through interacting with PpMYB114, potentiated the PpMYB114-mediated accumulation of anthocyanin in pear [156]. In the tea plant, UV-B irradiation-mediated bZIP1 upregulation leads to the promotion of flavonol biosynthesis by binding to the promoters of *MYB12*, *FLS*, and *UGT* and activating their expression; under shading, meanwhile, PIF3 inhibited flavonol accumulation by activating the expression of *MYB7*, which encodes a transcriptional repressor [157]. In peach, NAC1 was shown to regulate anthocyanin pigmentation through activating the transcription of *MYB10.1*, while NAC1 was repressed by SPL1 [158]. In the pear, PyWRKY26 interacts with PybHLH3 and activates the expression of PyMYB114, resulting in anthocyanin biosynthesis [159]. The BTB/TAZ protein MdBT2 represses anthocyanin biosynthesis, and MdGRF11 interacts with, and negatively regulates, MdBT2, leading to an increase in the expression of anthocyanin biosynthesis-related genes via the enhancement of the abundance of MdMYB1 protein [160]. SlBBX20 can bind the *SlDFR* promoter and directly activate its expression, which augments anthocyanin biosynthesis, while SlCSN5, a subunit of the COP9 signalosome, induces the degradation of SlBBX20 by enhancing its ubiquitination [161]. MdARF19 modulates anthocyanin biosynthesis by binding to the promoter of *MdLOB52* and further activating its expression [162]. BES1, a positive regulator in brassinosteroid signaling, inhibits the transcription of the MYB proteins MYB11, MYB12, and MYB111, thereby decreasing flavonol biosynthesis [163]

## 4. Perspectives

Flavonoids are abundantly present in land plants where they have diverse functions; as dietary components, they also exert a variety of beneficial effects in humans [2,16,164,165]. Elucidating the pathways involved in the biosynthesis of flavonoids will aid in better understanding their functions and potential uses. For example, the heterologous transformation of *F3′5′H* from *Campanula medium* (Canterbury bells) and *A3′5′GT* (*anthocyanin 3′,5′-O-glucosyltransferase* gene) from *Clitoria ternatea* (butterfly pea) driven by the native (*Chrysanthemum morifolium*) *F3H* promoter induced the synthesis of delphinidin and generated true blue Chrysanthemums [3,6,166]. Flavonoids have also been produced for food and medicine in engineered bacteria. The functional expression of plant-derived *F3H*, *FLS*, and *OMT* in *Corynebacterium glutamicum* yielded pterostilbene, kaempferol, and quercetin at high concentrations and purity [167]. In *Escherichia coli*, cyanidin 3-*O*-glucoside was generated through the induction of *ANS* and *3GT* using a bicistronic expression cassette [168]. These observations highlight the important application and economic value of deciphering the pathways involved in flavonoid biosynthesis.

Over the past few decades, flavonoid biosynthesis has been among the most intensively investigated secondary metabolic pathways in plant biology, and a considerable number of studies have contributed to revealing the exquisite mechanisms underlying the biosynthesis of flavonoids in plants [1,135]. However, several questions remain outstanding. For example, no comprehensive model exists as yet regarding which enzymes catalyze the formation of 3-deoxyanthocyanidin; additionally, the biosynthesis of phlobaphenes needs to be further improved. 

Plants are rich in diversity and often produce specific secondary metabolites. Recent studies have identified a unique flavone synthesis pathway in the root of the medicinal plant *S. baicalensis*, which generated root-specific flavones such as baicalein and norwogonin [68,70,71]. Accordingly, whether specific flavonoid biosynthesis pathways and metabolites also exist in other plants warrants further investigation, so as to continuously improve our knowledge of the flavonoid biosynthesis network.

In addition, combined multi-omics (genomics, transcriptomics, proteomics, and metabolomics) analysis provides a direction for the study of plant synthetic biology. In rice, a *flavonoid 7-O-glycosyltransferase* (*OsUGT706C2*) gene with a role in modulating flavonol (kaempferol) and flavone (luteolin and chrysoeriol) metabolism was identified by metabolite-based genome-wide association analysis [169]. Proteomics and transcriptomics, complemented with gas chromatography-mass spectrometry (GC-MS) analysis, aided in elucidating the flavonoid metabolic pathway during seed ripening in *Camellia oleifera* [170]. The constantly evolving multi-omics technology combined with big data analysis will likely lead to the identification of novel flavonoids and increased knowledge of the flavonoid biosynthesis network.

## Figures and Tables

**Figure 1 ijms-22-12824-f001:**
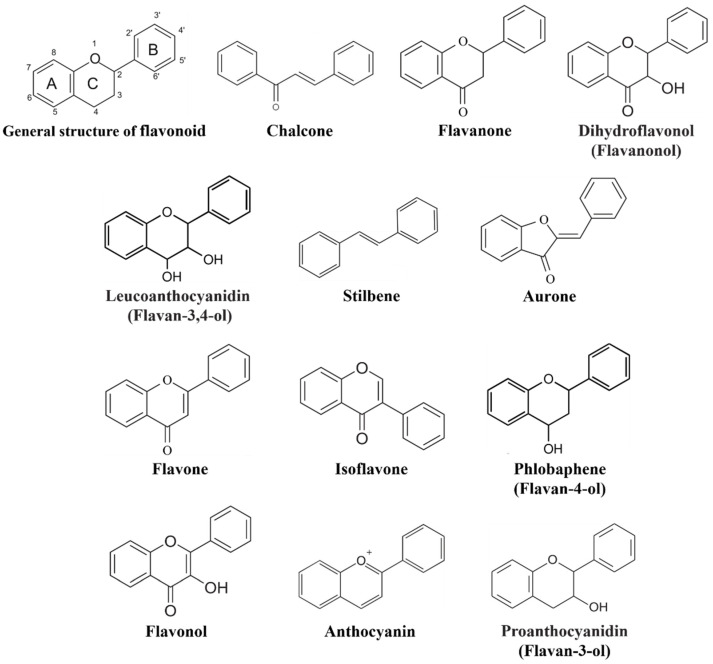
General structure of flavonoids.

**Figure 2 ijms-22-12824-f002:**
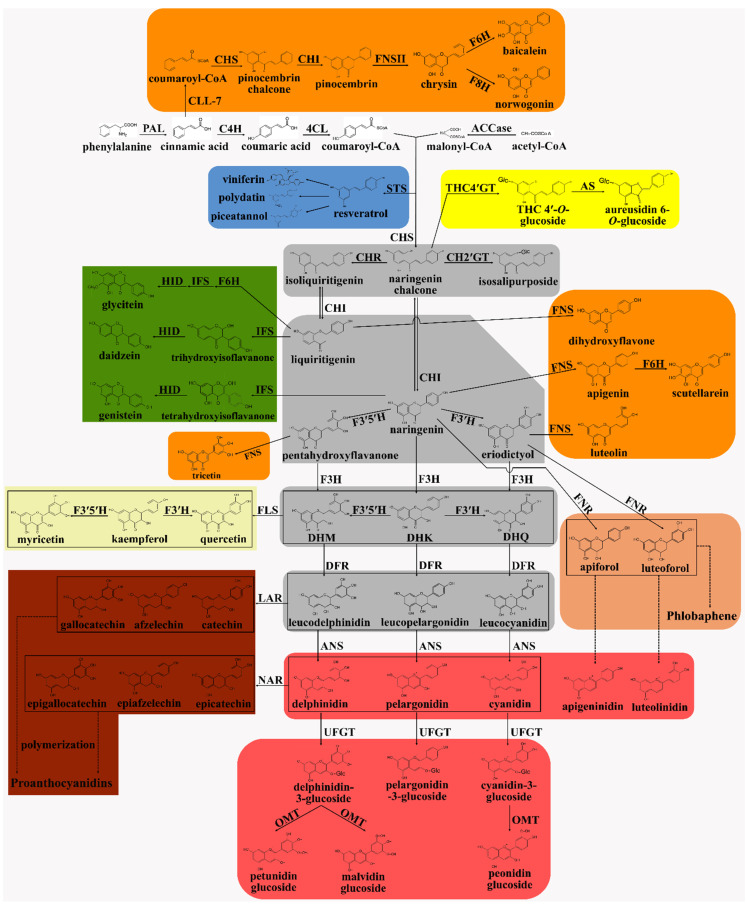
The flavonoid biosynthetic pathway in plants contains eight branches (represented by the eight colored boxes) and four important intermediate metabolites (gray boxes). The enzyme names and flavonoid compounds are abbreviated as follows: PAL, phenylalanine ammonia lyase; C4H, cinnamic acid 4-hydroxylase; 4CL, 4-coumarate: CoA ligase; ACCase, acetyl-CoA carboxylase; STS, stilbene synthase; CHS, chalcone synthase; CHR, chalcone reductase; CH2′GT, chalcone 2′-glucosyltransferase; CH4′GT, chalcone 4′-*O*-glucosyltransferase; AS, aureusidin synthase; CHI, chalcone isomerase; FNS, flavone synthase; CLL-7, cinnamate–CoA ligase; F6H, flavonoid 6-hydroxylase; F8H, flavonoid 8-hydroxylase; IFS, isoflavone synthase; HID, 2-hydroxyisoflavanone dehydratase; FNR, flavanone 4-reductase; F3H, flavanone 3-hydroxylase; F3′5′H, flavanone 3′,5′-hydroxylase; DHK, dihydrokaempferol; DHQ, dihydroquercetin; DHM, dihydromyricetin; FLS, flavonol synthase; DFR, dihydroflavonol 4-reductase; ANS, anthocyanidin synthase; UFGT, UDP-glucose flavonoid 3-*O*-glucosyltransferase; OMT, *O*-methyl transferases; LAR, leucoanthocyanidin reductase; ANR, anthocyanidin reductase.

**Figure 3 ijms-22-12824-f003:**
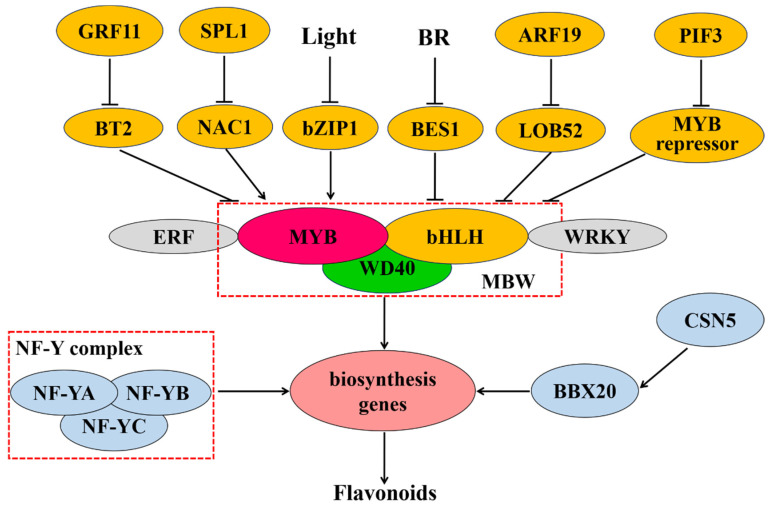
Transcriptional regulation of flavonoid biosynthesis in plants. Abbreviations are as follows: MYB, *v-myb* avian myeloblastosis viral oncogene homolog; bHLH, basic helix-loop-helix; NF-Y, nuclear factor Y; ERF, ethylene response factor; NAC, (NAM, ATAF, CUC); SPL, squamosa promoter binding protein-like; GRF, growth regulating factor; BT, BTB/TAZ; BBX, b-box protein; ARF, auxin response factor; LOB, lateral organ boundaries; BES1, BRI1-EMS-SUPPRESSOR 1; BR, brassinosteroid. The red dashed box represents the protein complex: MBW complex is constituted of three class of transcription factors (TFs), MYB, bHLH and WD40, while NF-Y complex is composed of TFs NF-YA, NF-YB, and NF-YC. TFs next to each other represent interaction of proteins.

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
