# Peer review of "The Flavonoid Biosynthesis Network in Plants"

_ijms, 2021, doi:10.3390/ijms222312824_

Round 1

Reviewer 1 Report

The authors wrote a review on the network of flavonoids biosynthesis in plants which could provide an overview of the process of flavonoids biosynthesis.

  1. Please rewrite the abstract, many grammar errors and weird expressions make it difficult to understand.
  2. I suggested the authors to rewrite the whole article since it’s really difficult for me to read and well-understand what the authors try to express.
  3. The resolution of Figure 1 is low, it's better to provide high resolution images.
  4. Figure 2 has the same problem as Figure 1, especially when the authors use the red background, I can barely see anything.
  5. It appears to me that the authors just listed a pile of results from published articles here, but the article lacks the logic connection. For example, why you listed the 8 branches in the article, what’s the relationship among them and what’s the importance in related research?

Author Response

  1. Please rewrite the abstract, many grammar errors and weird expressions make it difficult to understand.

Response: Thanks for your suggestion. We have sent the new revision to English native expert for the improvement.

  1. I suggested the authors to rewrite the whole article since it’s really difficult for me to read and well-understand what the authors try to express.

Response: Thank you very much for your advice of this manuscript. We have finished the revision based on the comments and have asked native-speaker English editing to polish the revision.

  1. The resolution of Figure 1 is low, it's better to provide high resolution images.

Response: Thank you very much. We have modified the Figure 1.

  1. Figure 2 has the same problem as Figure 1, especially when the authors use the red background, I can barely see anything.

Response: Thank you very much. We have modified the Figure 2.

  1. It appears to me that the authors just listed a pile of results from published articles here, but the article lacks the logic connection. For example, why you listed the 8 branches in the article, what’s the relationship among them and what’s the importance in related research?

Response: Thanks for your suggestion. Flavonoids are an important class of secondary metabolites widely found in plants, and have long been a major focus of intense research in plant biology. There were also some reviews of flavonoids, wrote by Winkel-Shirley (2001), Lepiniec et al. (2006), Tanaka et al. (2008), Dixon and Pasinetti (2010), Nabavi et al. (2018), Dong and Lin (2020) and so on, but they didn't list all the branches. Therefore, we focus on the flavonoid biosynthetic pathway in this review. We present a systematic summary of what is known of the flavonoid biosynthetic pathway in plants, presenting a model of flavonoid biosynthesis that includes eight branches (stilbene, aurone, flavone, isoflavone, flavonol, phlobaphene, proanthocyanidin, and anthocyanin biosynthesis) and four important intermediate metabolites (chalcone, flavanone, dihydroflavonol, and leucoanthocyanidin). Thank you very much for your suggestion, and we can take it as the next research.

Reviewer 2 Report

In this review, summarized the biosynthetic pathways of flavonoids and draw up an exhaustive map of flavonoids biosynthesis in plants, which contains eight branches (biosynthesis of stilbene, aurone, flavone, isoflavone, flavonol, phlobaphene, proanthocyanidin and anthocyanin) and four important intermediate metabolites (chalcone, flavanone, dihydroflavonol and leucoanthocyanidin). The manuscript is well structured and well discussed. However, some points should be checked and corrected.  Therefore, I recommend major revision according to given my comments.

  • Please add figure of molecular mechanism of flavonoids biosynthesis.
  • The MS English needs to be improved. The article's English must be carefully checked for grammatical errors.
  • In Conclusion, the authors should add the significance of this review, and its potential practical application.

Author Response

  • Please add figure of molecular mechanism of flavonoids biosynthesis.

Response: Thanks for your suggestion. We have added the content ‘3. Transcriptional regulation of flavonoid biosynthesis in plants’ and ‘Figure 3. Transcriptional regulation of flavonoid biosynthesis in plants’.

  • The MS English needs to be improved. The article's English must be carefully checked for grammatical errors.

Response: Thank you very much. We have finished the revision based on the comments and have asked native-speaker English editing to polish the revision.

  • In Conclusion, the authors should add the significance of this review, and its potential practical application.

Response: Thank you very much for your advice. We have added the significance and its potential application in the first paragraph of ‘Perspectives’.

Reviewer 3 Report

The review by Liu et al. proposes an interesting presentation of the state of the art knowledge about the biosynthesis of flavonoids in plants. Their approach appears simple and can be valuable by many researchers to have a general guide for a first understanding of this category of specialized metabolism. However, there are some major and minor flaws that hamper the readability and the strength of the manuscript and should be addressed before it can be considered for publication.

Major flaws:

  • The authors provide a good description of the biosynthetic pathways of flavonoids presenting the structural element of the pathways. However, the importance of regulatory elements in the biosynthesis of these specialized metabolites should not be forgotten. For example, even if the authors cited many previous articles dealing with the important roles of R2R3-MYB transcription factors in flavonoids biosynthesis this aspect is not treated in the manuscript. This should be more emphasized with the latest research findings considering both model and non-model organisms.

Minor flaws:

  • The writing is good but some crucial information is overlooked or their logical sequence should be followed. Such as in the first lane the authors refer to flavonoids as pigments stored in cell vacuoles. I think that a more broad description of the role of these molecules (as in the protection of plants against damage from biotic and abiotic stress for instance) and their origin should be given before telling in which compartment they are located in the cell.

Author Response

  • The authors provide a good description of the biosynthetic pathways of flavonoids presenting the structural element of the pathways. However, the importance of regulatory elements in the biosynthesis of these specialized metabolites should not be forgotten. For example, even if the authors cited many previous articles dealing with the important roles of R2R3-MYB transcription factors in flavonoids biosynthesis this aspect is not treated in the manuscript. This should be more emphasized with the latest research findings considering both model and non-model organisms.

Response: Thank you very much for your advice of this manuscript. We have added the content ‘3. Transcriptional regulation of flavonoid biosynthesis in plants’.

Minor flaws:

  • The writing is good but some crucial information is overlooked or their logical sequence should be followed. Such as in the first lane the authors refer to flavonoids as pigments stored in cell vacuoles. I think that a more broad description of the role of these molecules (as in the protection of plants against damage from biotic and abiotic stress for instance) and their origin should be given before telling in which compartment they are located in the cell.

Response: Thanks for your suggestion. We have revised the description of this review, such as the first lane ‘Flavonoids comprise a group of phenylpropanoids that as water-soluble pigments stored in the vacuoles of plant cells (Dong and Lin, 2020)’. And we have sent the new revision to English native expert for the improvement.

Round 2

Reviewer 1 Report

I'm satisfied with authors' answers. The authors still need to add the figure caption and legend for the last figure they added in the revised version.

Author Response

Response: Thank you very much for your advice of this manuscript. We have added the figure caption and legend in the Figure 3 (line 443-451).

Reviewer 2 Report

Requested corrections were completed.

Author Response

Response: Thank you very much for your review.

Reviewer 3 Report

The new version is highly improved.

Author Response

(The authors gave the same response as above.)
